# New Genotype G3 P[8] of Rotavirus Identified in a Mexican Gastroenteric Rabbit

**DOI:** 10.3390/v16111729

**Published:** 2024-11-02

**Authors:** Emmanuel Reynoso-Utrera, Linda Guiliana Bautista-Gómez, Salvador Fonseca-Coronado, Juan Diego Pérez-de la Rosa, Valeria Jazmín Rodríguez-Villavicencio, Camilo Romero-Núñez, Ariadna Flores-Ortega, Pedro Abel Hernández-García, José Simón Martínez-Castañeda

**Affiliations:** 1Laboratorio de Biotecnología, Biología Molecular y Genética, Centro Universitario UAEM Amecameca, Universidad Autónoma del Estado de México, Amecameca 56900, Mexico; mvz.manolo77@gmail.com (E.R.-U.); rodriguez_villavicencio@hotmail.com (V.J.R.-V.); ariadnafloresortega@gmail.com (A.F.-O.); 2Laboratorio de Investigación en Inmunología y Salud Pública, Facultad de Estudios Superiores Cuautitlán, Universidad Nacional Autónoma de México, San Sebastian Xhala 54714, Mexico; fonsecacoronado@yahoo.com; 3Laboratorio de Diagnóstico Molecular JGG & RNL, Facultad de Medicina Veterinaria y Zootecnia, Universidad Autónoma Benito Juárez de Oaxaca, Oaxaca 68120, Mexico; juan.perez@senasica.gob.mx; 4Hospital Veterinario DERMAVET, Ciudad de México 09510, Mexico; mvzcamilo@yahoo.com.mx; 5Laboratorio Multidisciplinario en Investigación, Centro Universitario UAEM Amecameca, Universidad Autónoma del Estado de México, Amecameca 56900, Mexico; pahernandezg@uaemex.mx; 6Centro de Investigación y Estudios Avanzados en Salud Animal, Facultad de Medicina Veterinaria y Zootecnia, Universidad Autónoma del Estado de México, Toluca 50000, Mexico

**Keywords:** rotavirus, rabbit, genotypic characterization, G3P[8], Mexico

## Abstract

Rotavirus species A (RVA) is a major cause of acute viral gastroenteritis in young humans and diverse animal species. The study of the genetic characteristics of RVAs that infect rabbits (Oryctolagus cuniculus) (lapine strain [LRV]) has been limited, and, to date, the most common and epidemiologically important combinations of G/P genotypes in rabbits have been reported to be G3 P[14] and G3 P[22]. In this study, a rotavirus species A detected from an outbreak of enteritis in a Mexican commercial rabbitry was genotypically characterized. Based on sequence and phylogenetic analysis of the VP7 and VP4 genes, the strain identified in this study (C-3/15) demonstrated a G3 P[8] genotype of rotavirus, which had not previously been reported in rabbits. Moreover, both genes were closely related to human, not lapine, rotaviruses. The G3 genotype has been reported in a wide variety of hosts, including humans and rabbits, whereas the P[8] genotype has only been reported in humans. Because this combination of genotypes has never been identified in rabbits, it is proposed that the finding presented here is possibly the result of an interspecies transmission event. This is the first work to study the molecular characteristics of rotaviruses in rabbits in Mexico, as well as the identification of human G3 and P[8] genotypes in a rabbit with enteric disease.

## 1. Introduction

Species A rotaviruses (RVAs), members of the realm *Riboviria*, the family *Sedoreoviridae*, and the genus Rotavirus [1], are the main cause of gastroenteritis in young humans and several animal species worldwide [2,3], including rabbits [4,5]. Enteric diseases have an important role in breeding farms, as they cause serious economic losses due to mortality, growth depression, and a decline in the conversion rate [5,6,7].

The rotavirus genome consists of 11 segments of double-stranded RNA (dsRNA), wrapped in a triple-layer protein capsid and spikes projecting to the surface. The genome segments encode six structural viral proteins (VP1–VP4, VP6, and VP7) and six non-structural viral proteins (NSP1–NSP6) [8,9]. Because of the segmented nature of the RV genome, its genetic diversity can be generated through genome reassortment and rearrangement events involving one or more segments [10]. However, although it has several evolutionary mechanisms, the main ones appear to be frequent point mutations in all RNA segments, which may occur sporadically or accumulate sequentially [11], generating continuously due to the high error rate of the RV RdRp [12] and genome reassortments.

According to the serological reactivity and genetic variability of the VP6 intermediate layer protein, nine species of rotavirus have been differentiated (A, B, C, D, F, G, H, I, and J), differing in their antigenic and epidemiological characteristics, animal species, and the most susceptible age groups [1,13,14].

Based on the antigenic specificity of VP6, four subgroups (I, II, I + II, and noI/noII) have been distinguished within species A, which are assigned as a function of the presence or absence of two different reactive epitopes with one, both, or none of the monoclonal antibodies [15,16]. VP7 (glycoprotein) and VP4 (protease-sensitive protein) are the proteins of the outer layer that represent the major antigenic determinants because they independently elicit neutralizing antibody responses, and it is their genetic/antigenic variability that determines the G and P genotypes, respectively [2,8]. At least 36 G and 51 P genotypes have been recognized to date [1,17].

Throughout the world, various combinations of G and P genotypes of RV have been shown to be relatively common to a particular host species. For example, the most typical combinations found in humans are G1P[8], G3P[8], G4P[8], G9P[8], G12P[8], and G2P[4], with G1P[8] being the most prevalent worldwide [18,19,20]. In animals, the most common genotype combinations are G6, G8, and G10 with P[1], P[5], and P[11] in cattle; G3–G5, G9, and G11 with P[6] and P[7] in pigs [6,7]; G3 and G14 with P[12] in horses [21]; G3 with P[3] and P[9] in dogs and cats [22]; and G3 with P[14] and P[22] [5,23,24,25,26,27] in rabbits.

Since RVA was recognized as a major cause of viral diarrhea in young mammals, the genetic diversity of their VP7 and VP4 genes has been extensively studied in humans and some mammals [20,28], particularly in livestock animals [7]. However, among these species, rabbits have not been considered; in our country, the role of rotavirus in enteric diseases in this species is unknown, as well as the economic impact that can be generated. Currently, few studies have investigated RV infection in rabbits in general and their molecular characteristics in particular. Thus, there are few isolated and partially characterized lapine strains [29]. In Mexico, the presence of rotavirus in rabbits has recently been reported [30,31]. However, the strain and its genotypic characteristics have not yet been determined. The aim of this study was to characterize the rotavirus genotype in Mexican rabbits through molecular analysis of the VP4 and VP7 genes.

## 2. Materials and Methods

### 2.1. Sample Collection

From April 2015 to July 2020, an outbreak of diarrhea was reported on a rabbit production farm from the eastern region of Mexico State; the major farm had a total of 3000 animals in production. The affected animals were young rabbits of 40–60 days old. The rabbits that died at the time of the farm visit and those that had diarrhea were transferred to the amphitheater of the UAEM Amecameca University Center of the Autonomous University of the State of Mexico, whereas the live animals were humanely slaughtered according to the NOM-033-SAG/ZOO-2014.

Rabbits were examined postmortem for tissue sampling; intestinal contents and duodenum samples (0.5 × 2.0 cm) were stored in 2 mL Eppendorf microtubes with RNAlater solution as the stabilizer and then were labeled and frozen at −80 °C until RNA extraction.

### 2.2. RNA Extraction and Virus Detection

Viral RNA was obtained from both tissue (0.5 mm) and intestinal contents (200 μL) using the GeneJET Viral DNA and RNA Purification kit (Thermo Scientific^TM^, USA), according to the manufacturer’s instructions; for each sample, 200 μL of Lysis Solution (which contained guanidine hydrochloride supplemented with Carrier RNA) and 50 μL of Proteinase K was added. Next, each sample was incubated for 15 min at 56 °C in a thermomixer and processed by spin columns. Once the viral RNA was obtained, a single-step RT-PCR was performed using the commercial kit SuperScript^®^ III One Step RT-PCR with Platinum^®^ Taq (Invitrogen^TM^, Carlsbad, CA). In a final volume of 50 μL, they were placed with 25 μL of 2X Reaction Mix (a buffer containing 0.4 mM of each dNTP and 3.2 mM MgSO_4_), 1 μg of Template RNA, 10 μM of sense and anti-sense primer, and 2 μL of SuperScript™ III RT/Platinum™ Taq Mix. The virological examination consisted of the detection of the VP4 and VP7 genes of rotavirus through the reverse polymerase chain reaction (RT-PCR), from the total RNA extracted from the sample. Primers reported by Gentsch et al. [32] were used for the amplification of an 876 bp fragment that encodes the VP4 protein (Con 3 [nt 11–32] F 5′ TGGCTTCGCCATTTTATAGACA 3′ and Con 2 [nt 887–868] R 5′ ATTTCGGACCATTTATAACC 3′). For VP7 amplification, the primers reported by Gouvea et al. [33] were used (Beg 9 [nt 1–28] F 5′ GGCTTTAAAAGAGAGAATTTCCGTCTGG 3′ and End 9 [nt 1062–1036] R 5′ GGTCACATCATACAATTCTAATCTAAG 3′). RotaTeq^®^ vaccine was used as a positive control and as a negative control; RNA extracted from blood and tissue samples from rabbits without clinical signs of gastrointestinal disease, such as diarrhea, dehydration, or anorexia, and whose feces were evaluated and found to be negative for rotavirus were considered in this group. PCR products were visualized on 2% agarose gels stained with ethidium bromide.

### 2.3. Nucleotide Sequencing and Phylogenetic Analyses

Gel-purified PCR fragments were directly sequenced in an ABI Prism 3100 Genetic Analyser (Applied Biosystems, Foster City, CA, USA). The obtained sequence data were analyzed using the basic local alignment search tool of the National Center for Biotechnology Information database. Phylogenetic and molecular evolutionary analyses were conducted using the MEGA X software [34]. To construct the phylogenetic trees, additional sequences were obtained from GenBank (http://www.ncbi.nlm.nih.gov, accessed on 11 June 2020). As a complementary tool for the assignment of genotypes, the RotaC 2.0 automatic genotyping program for rotavirus group A (available at http://rotac.regatools.be/, accessed on 11 June 2020) [35] was used.

### 2.4. Ethics Statements

This study was authorized by the Bioethics Committee of the Centro Universitario UAEM Amecameca (CBE/13/2014).

### 2.5. GenBank Accession Numbers

Partial sequences of the VP7 and VP4 genes of the rotavirus strain C-3/15 were deposited in GenBank under the accession numbers MT267356 and MT267357.

## 3. Results

A total of 188 rabbit samples with enteric signs were analyzed, and rotavirus was detected in 6 of them. The positive sample (C-3/15) came from an animal of approximately 50 days of age, which presented signs of diarrhea, abdominal distension, dehydration, and anorexia. The characterization of the sample revealed that the VP7 and VP4 genes corresponded to the G3 P[8] genotypes, whereas none were closely related to LRV.


**VP4 analysis and determination of P genotype**


The partial nucleotide sequence (767 bp) and the deduced amino acid sequence (215 aa) of the gene that encodes the VP4 sequence of strain C-3/15 were determined. Sequence comparison indicated that the VP4 sequence of strain C-3/15 was most closely related to the human rotavirus strain RVA/Human-wt/USA/VU12-13-42/2013/G12P[8] (97.63% nt and 99.04% aa identity). Furthermore, they shared identities of 94.62–97.50% at the nucleotide level and 94.71–97.12% at the amino acid level only with human rotavirus genotype P[8].

In order to study the relationship between C-3/15 and other known RVA strains, a phylogenetic tree was constructed by the neighbor joining method, where different representative P genotypes of various rotavirus host species were also included (Figure 1).

Through the automatic genotyping tool for RVA RotaC 2.0, the VP4 sequence of C-3/15 showed an identity of 95.6% with rotavirus P[8] isolated from a human.


**VP7 analysis and determination of G genotype**


The VP7 partial nucleotide sequence of the strain C-3/15 was 849 bp in length, whereas the deduced amino acid sequence was 269 aa. Sequence comparison indicated that the sequence obtained was more closely related to the human rotavirus strain 7177-1042 [36], with genotype G4 (97.64% nt and 98.14% aa identity), and shared identities of 90.69–96.81% at the nucleotide level with human rotavirus strains with G3 genotypes.

On the other hand, the VP7 sequence of the C-3/15 strain shared identities of 86.45, 85.98, and 85.87% with the strains 30/96 [27], 3489/3 [23], and 308/01 [27], respectively (Table 1).

Consistent with the sequence identity data, a neighbor joining tree was constructed to observe the phylogenetic relationships of the VP7 sequence of the C-3/15 strain with other RVA strains, including the LRV strains from the G3 genotype (Figure 2).

Due to the important relationship observed between the C-3/15 VP7 sequence with the 7177-1042 strain genotyped as G4 and considering that all other related strains corresponded to the G3 genotype, a second VP7 phylogenetic tree was constructed, which included sequences of different G genotypes (G1–G13) of RVA (Figure 3).

Through the RotaC 2.0 tool, the VP7 sequence of C-3/15 was associated with the VP7 gene (86.4%) of the lapine strain RVA/Rabbit-tc/ITA/30-96/1996/G3P[14] [27].

Finally, considering the sequences of both VP4 and VP7 genes of C-3/15, a phylogenetic tree was created with the objective of observing the relationships of G and P genotypes of different RVA strains (Figure 4).

## 4. Discussion

Enteric diseases have an important role in rabbit production, as they can lead to severe economic losses due to mortality, growth depression, and a reduced feed conversion rate. Enteric syndrome is one of the most important diseases in rabbits, especially in relation to its productive and economic impact. Among the different pathogens that can be found in rabbits with enteric disease, viruses seem to have an important but not definitive role [37]. Bacteriological and coproparasitoscopic tests identified the presence of two potential pathogens in rabbits [38]: *E. coli* in three samples and *Eimeria* spp. in all the samples analyzed. However, rotavirus was identified in only six of them, suggesting that the role of rotavirus in rabbits with enteric disease is not determinant of the disease presentation. The findings presented here reinforce the hypothesis that infectious enteritis in rabbits is of multifactorial origin, with different pathogens acting synergistically to induce gastroenteritis [5,23,38,39].

Rotavirus lapine is grouped within species A, subgroup I, and has been identified in Canada (LRV) [40]; Japan (R-2) [41]; Italy (82/311F) [42], (30/96, 160/01, 229/01, 308/01) [27], and (30/96) [43]; the United States (ALA, C-11, BAP, BAP-2) [24,44]; Hungary (229-01, 3489/3) [23]; China (N-5) [26]; the Netherlands (K1130027) [45]; and Korea (13D025) [46] and (Rab1404) [47]. Those that have been characterized belong mostly to the G3 genotype of VP7 and the P[14] and the P[22] genotype of VP4.

The VP7 sequence of C-3/15 was initially associated with the G4 genotype of strain 7177-1042 (Figure 2). However, in subsequent analyses, it was confirmed that C-3/15 belongs to the G3 genotype (Figure 3).

Analysis of the few strains identified has revealed a substantial antigenic/genetic homogeneity of LRVs [5,25,48]. However, based on the sequences and phylogenetic analyses of the VP4 (Figure 1) and VP7 (Figure 3) genes, the genotype found in this study, identified as G3 P[8], contrasts with the RV genotypes reported in rabbits worldwide. The G3 RVA genotype has been described in a wide range of different host species, including humans and rabbits—whereas the P[8] genotype is, in addition to being the most common genotype, only detected in human rotavirus—and, unusually, in pigs in Spain [49].

Throughout the world, various combinations of RV genotypes P and G have been shown to be relatively common to a particular host species. However, numerous unusual or exotic genotypes have also been reported and denominated in this way to be identified in hosts different from those that are commonly affected or to be found in different combinations of genotypes in different host species. In most cases, these genotypes are derived from strains common to other hosts, possibly through interspecies transmission coupled with reassortment events [50]. Kim et al. [46] reported evidence of such events in rabbits when performing a study that identified the G3 P[3] genotype of RV from a small outbreak of enteritis in a Korean rabbitry. In this study, the VP4 sequence was found to be closely related to the Rhesus simian rotavirus strain (RRV) with the P[3] genotype, which had previously been detected [51] in goats in the same country. They concluded that although the rabbit production farm from which RV was detected was far from other livestock farms, the heterologous RRV virus could productively replicate and spread horizontally to the rabbit population; in addition, phylogenetic analysis can predict the epidemiological origin of the viral strains involved.

Because both C-3/15 VP4 and VP7 sequences showed a close relationship with human rotavirus but not with LRV (Figure 4), it is proposed that the identification of rotavirus in a rabbit sample of intestinal tissue is possibly related to an event of interspecies transmission, where the rabbit was in contact with viral particles through contaminated food or water, possibly involving poor management or inadequate sanitary measures. However, it is not possible to determine the participation of RV in the generation of enteric markers of disease. Still, it is surprising to detect a rotavirus of human origin in a rabbit with diarrheal disease. This event, as far as we know, had not previously been documented. The possibility that the finding presented here is the result of genetic reassortments between human rotavirus strains or human and lapine strains should also be considered, for which future research is suggested.

This is the first work that studied the molecular characteristics of rotavirus in rabbits in Mexico, as well the identification of human G3 and P[8] genotypes in a rabbit with diarrheal disease.

## Figures and Tables

**Figure 1 viruses-16-01729-f001:**
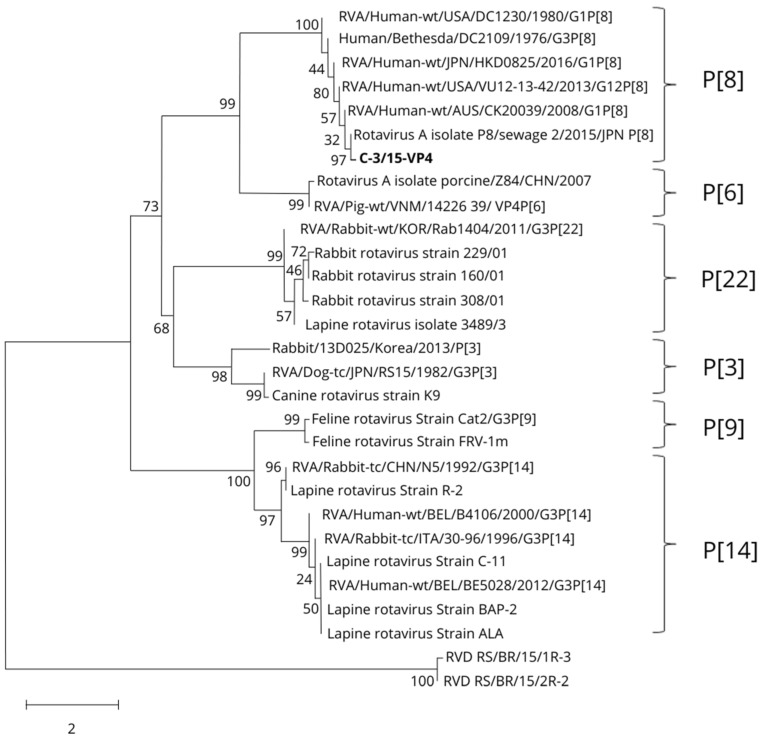
Phylogenetic tree of the VP4 nucleotide sequence based on the neighbor joining method with a bootstrap of 1000 replicates, showing the sequence associations of different P genotypes of RV. C-3/15-VP4 is reported in this study.

**Figure 2 viruses-16-01729-f002:**
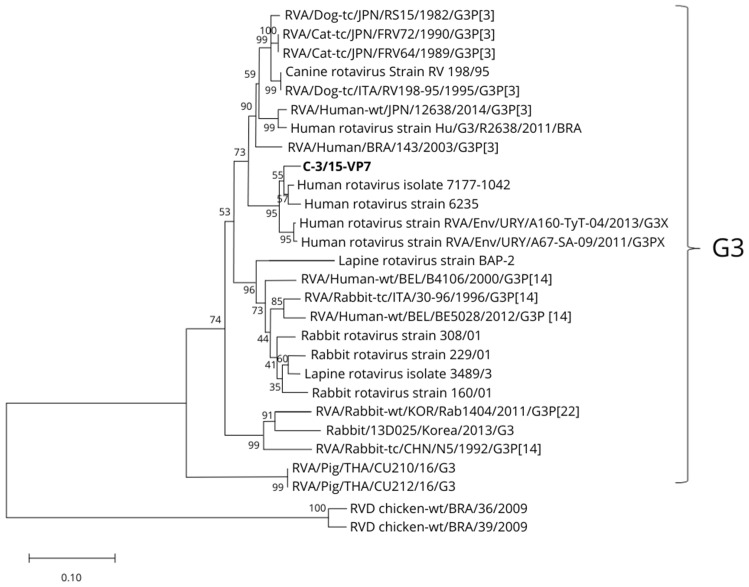
Phylogenetic tree of VP7 nucleotide sequence based on the neighbor joining method with a bootstrap of 1000 replicates, where the relationships between the C-3/15-VP7 strain and RVA strains with G3 genotype are shown.

**Figure 3 viruses-16-01729-f003:**
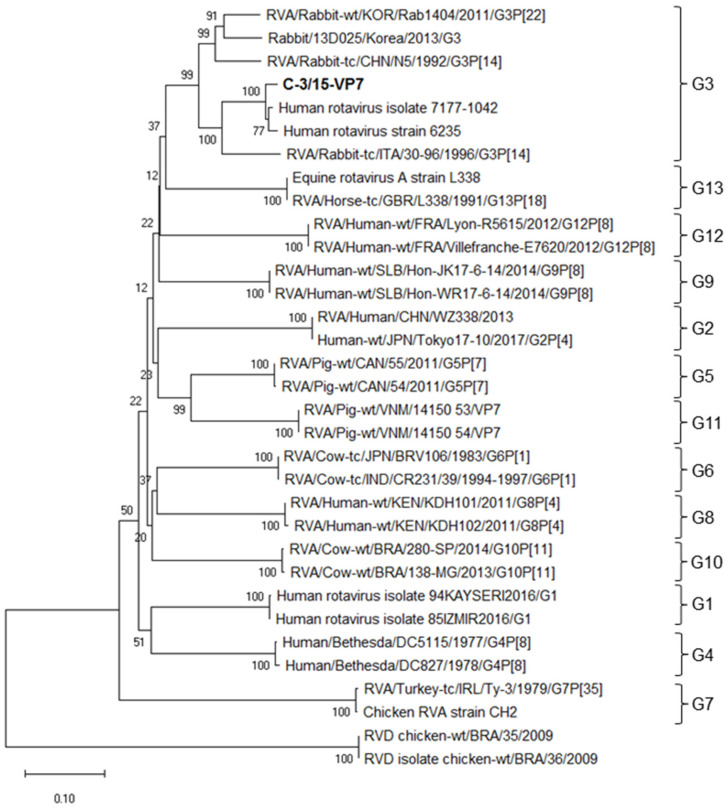
Phylogenetic tree of VP7 nucleotide sequence based on the neighbor joining method with a bootstrap of 1000 replicates, showing the associations of different G genotypes of RVA. C-3/15-VP7 is reported in this study.

**Figure 4 viruses-16-01729-f004:**
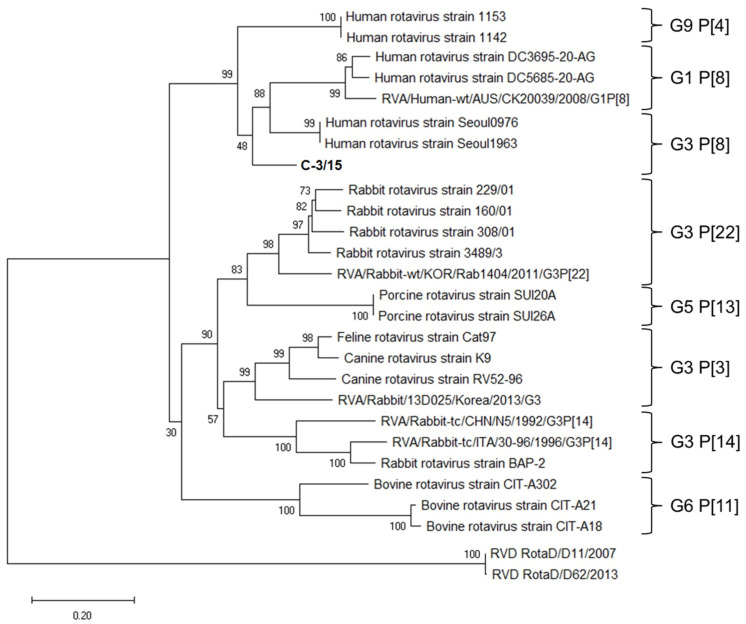
Phylogenetic tree of VP7 and VP4 nucleotide sequences, by the neighbor joining method with a bootstrap of 1000 replicates. The relationships of P and G genotypes of different strains of RVA are shown. C-3/15 is reported in this study.

**Table 1 viruses-16-01729-t001:** Rotavirus strains that were most closely related to VP7 strain C-3/15.

Strain	Origin	Genotype	Identity nt (%)	Identity aa (%)
HRV 7177-1042	Human	G4	97.64	98.14
HRV 6235	Human	G3	96.81	97.03
RVA/Env/URY/A160-TyT-04/2013/G3PX	Sewage	G3	97.03	97.98
RVA/Env/URY/A67-SA-09/2011/G3PX	Sewage	G3	96.76	97.98
HRV Ro1845	Human	G3	90.69	95.54
RVA/Dog-tc/JPN/RS15/1982/G3P[3]	Canine	G3	90.58	95.54
RVA/Cat-tc/JPN/FRV72/1990/G3P[3]	Feline	G3	90.34	95.54
RVA H-2 VP7	Equine	G3	87.47	92.94
RVA/Rabbit-tc/ITA/30-96/1996/G3P[14]	Rabbit	G3	86.45	92.57
LRV 3489/3	Rabbit	G3	85.98	91.82
LRV 308/01	Rabbit	G3	85.87	90.71

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
