# Peer review of "New Genotype G3 P[8] of Rotavirus Identified in a Mexican Gastroenteric Rabbit"

_viruses, 2024, doi:10.3390/v16111729_

Round 1

Reviewer 1 Report

Comments and Suggestions for Authors

The manuscript "New genotype G3 P[8] of rotavirus identified in a Mexican gastroenteric rabbit" by Emmanuel Reynoso-Utrera et al. reports the discovery of a rotavirus strain with a G3 P[8] genotype in rabbits, previously known only in humans and pigs. This finding from a Mexican rabbitry suggests potential interspecies transmission and highlights the genetic diversity of rotavirus. The study underscores the need for further investigation into the mechanisms of rotavirus spread among different species and its implications for animal health management.

1. The manuscript provides a comprehensive genotypic characterization of the rotavirus strain. However, could the authors provide more details on the RT-PCR  experiments, for example what are the concentrations of the reagents in “a 200 μL of Lysis Solution (supplemented with Carrier RNA), and 50 μL of Proteinase K”. 

2. Could the authors provide more details on the controls used during the RT-PCR to rule out contamination?

3. The authors need to provide more detail information on how the rabbits were monitored for symptoms and how differential diagnoses were ruled out in establishing rotavirus as the causative agent of the outbreak.

4. Given that only six samples tested positive for rotavirus out of 188 rabbit samples, how do the authors address the potential bias in their sampling method and its impact on the conclusions about genetic diversity and interspecies transmission?

5. Could the authors provide additional details about the sensitivity and specificity of the assays used to detect and genotype the rotavirus strains? How might these factors influence the interpretation of the results.

6. Could the authors elaborate on the specific environmental or management practices at the affected rabbitry that might have contributed to the outbreak?

Author Response

  1. The manuscript provides a comprehensive genotypic characterization of the rotavirus strain. However, could the authors provide more details on the RT-PCR  experiments, for example what are the concentrations of the reagents in “a 200 μL of Lysis Solution (supplemented with Carrier RNA), and 50 μL of Proteinase K”. 

Regarding the concentration of reagents for viral RNA extraction, the kit used does not indicate the reagent concentrations, as can be found in the user manual (https://assets.thermofisher.com/TFS-Assets/LSG/manuals/MAN0012669_GeneJET_Viral_DNA_RNA_Purification_UG.pdf), for this reason they were not included in the manuscript. However, the content of guanidine hydrochloride in the lysis solution has been added, as indicated by the manufacturer, and the sample quantities used have also been included: 200ul for intestinal content and 0.5mm of tissue.

As for RT-PCR, a breakdown of the reagents used has been included, as indicated by the manufacturer: SuperScript® III One Step RT-PCR with Platinum® Taq (InvitrogenTM). In a final volume of 50 μL, they were placed 25 μL of 2X Reaction Mix (a buffer containing 0.4 mM of each dNTP, 3.2 mM MgSO4 ), 1 μg of Template RNA, 10 μM of sense and anti-sense primer, and 2 μL of SuperScript™ III RT/Platinum™ Taq Mix.

  1. Could the authors provide more details on the controls used during the RT-PCR to rule out contamination?

The pentavalent RotaTeq vaccine was used as a positive control, and as a negative control, RNA extracted from blood and tissue samples from rabbits without clinical signs of gastrointestinal disease, such as diarrhea, dehydration or anorexia, whose feces were evaluated and found to be negative for rotavirus were considered in this group.

It is important to mention that, in order to avoid any type of cross contamination, the extraction of RNA from the vaccine was carried out independently of the processing of the samples obtained in the field, however a negative control sample was always processed together with the samples to validate that there was no contamination and the work area was disinfected with RNaseZap™, as well as the micropipettes, after which the material was sterilized for 60 min in a hood with ultraviolet light, this procedure is routinely performed as part of the workflow for sample processing.

  1. The authors need to provide more detail information on how the rabbits were monitored for symptoms and how differential diagnoses were ruled out in establishing rotavirus as the causative agent of the outbreak.

In the article presented, we agree with various reports worldwide in which it is mentioned that the role of rotavirus is not clearly identified in the causes of gastroenteritis in rabbits, mainly because, in rabbits it is known as multifactorial syndrome in which bacteria and parasites of the genus Eimeria are present, in this sense the role of viruses has been discussed on whether they facilitate the colonization of other pathogens by damaging the epithelium of the intestine, however it has not been possible to conclude, nor has it been possible to identify which pathogen colonizes first, that is why as a conclusion we mention "it is not possible to determine the participation of RV in the generation of enteric signology"

However, the present research aims to visualize the zoosanitary status of Rabbit Farming in Mexico, since it is a forgotten and unknown activity, by presenting the circulating pathogens, we want to visualize a problem that could become a public health problem in Mexico, since rabbit farming is an activity that is carried out mainly as a backyard activity, where the members of the family are Those in charge of animal care and also the government have tried to promote the activity as a way to combat poverty and extreme poverty in some localities, however there are no monitoring programs that allow for adequate control of circulating pathogens, so, when a problem becomes evident, we hope that the activity will be supported in the future by government agencies to support producers and society in general.

  1. Given that only six samples tested positive for rotavirus out of 188 rabbit samples, how do the authors address the potential bias in their sampling method and its impact on the conclusions about genetic diversity and interspecies transmission?

Several studies worldwide have reported the identification of new rotavirus genotypes in various species with the sequencing of only one sample, and in the same way the identification of rotavirus in percentages less than 5% has been mentioned, to mention one, in the study reported by Bwogi, et. Al. 2023, out of a total of 418 goats sampled, they found the presence of rotavirus in 9 of them, indicating a similarity with the results presented in this study, in the same way it has happened in studies in equines, guanacos, pigs, cattle and rabbits, some of the studies cited are:

  • Bányai K, Forgach P, Erdelyi K, Martella V, Bogdan A, Hocsak E, et al. Identification of the novel lapine rotavirus genotype P[22] from an outbreak of enteritis in a Hungarian rabbitry. Virus Res. 2005, 113(2), 73-80.
  • Browning GF, Chalmers RM, Fitzgerald TA, Snodgrass DR. Serological and genomic characterization of L338, a novel equine group A rotavirus G serotype. J Gen Virol. 1991;72 ( Pt 5):1059-1064. doi:10.1099/0022-1317-72-5-1059
  • Bwogi, J., Karamagi, C., Byarugaba, D. K., Tushabe, P., Kiguli, S., Namuwulya, P., Malamba, S. S., Jere, K. C., Desselberger, U., & Iturriza-Gomara, M. (2023). Co-Surveillance of Rotaviruses in Humans and Domestic Animals in Central Uganda Reveals Circulation of Wide Genotype Diversity in the Animals. Viruses15(3), 738.
  • Ghosh S, Kobayashi N, Nagashima S, et al. Full genomic analysis and possible origin of a porcine G12 rotavirus strain RU172. Virus Genes. 2010;40(3):382-388. doi:10.1007/s11262-010-0454-y
  • Guo, D., Liu, J., Lu, Y., Sun, Y., Yuan, D., Jiang, Q., … Qu, L. (2012). Full genomic analysis of rabbit rotavirus G3P[14] strain N5 in China: Identification of a novel VP6 genotype. Infection, Genetics and Evolution, 12(7), 1567–1576.
  • Khamrin P, Maneekarn N, Peerakome S, Yagyu F, Okitsu S, Ushijima H. Molecular characterization of a rare G3P[3] human rotavirus reassortant strain reveals evidence for multiple human-animal interspecies transmissions. J Med Virol. 2006;78(7):986-994. doi:10.1002/jmv.20651
  • Lavazza A, Capucci L, editors. Viral infection of rabbits. Proceedings of the 9th World Rabbit Congress, Verona, Italy, 10-13 Jun 2008.
  • avazza A, Cerioli M, Martella V, Tittarelli C, Grilli G, Brivio R, et al. editors. Rotavirus in diarrheic rabbits: prevalence and characterization of strains in Italian Farms. Proceedings of the 9th World Rabbit Congress, Verona, Italy, 10-11 June 2008.
  • Licois, D, editor. Domestic rabbit enteropathies. Proceedings of the 8th World Rabbit Congress; 2004 Sep 7-10; 2004. Puebla, Mexico. 385-403.
  • Nieddu D, Grilli G, Gelmetti D, Gallazzi D, Toccaciel S, Lavazza A, editors. Electron microscopy detection of viral agents in rabbits with enteropathy during the period 1982-1999 in Italy. Proceedings of the 7th World Rabbit Congress; 2000 July. Valencia, Spain. 4-7.
  • Parreño V, Bok K, Fernandez F, Gomez J. Molecular characterization of the first isolation of rotavirus in guanacos (Lama guanicoe). Arch Virol. 2004;149(12):2465-2471. doi:10.1007/s00705-004-0371-2
  • Perkins C, Mijatovic-Rustempasic S, Ward ML, Cortese MM, Bowen MD. Genomic Characterization of the First Equine-Like G3P[8] Rotavirus Strain Detected in the United States. Genome Announc. 2017;5(47):e01341-17. Published 2017 Nov 22. doi:10.1128/genomeA.01341-17

  1. Could the authors provide additional details about the sensitivity and specificity of the assays used to detect and genotype the rotavirus strains? How might these factors influence the interpretation of the results.

The methodology used in this study is the same as that reported worldwide for the identification of rotavirus in animals and humans, including the initiators reported by Gentsch et al. 1992 And Gouvea et al.1990, as reported in the Manual of rotavirus detection and characterization methods, published by the World Health Organization, however, as stated in the article, it is possible that the genetic diversity of rotavirus is greater than that reported and the development of new primers designed from consensus sequences specific to each species is necessary, however, as mentioned in this study, it would be an argument to continue with the identification of rotavirus in rabbits in Mexico, taking advantage of the questioning we want to comment that, within the research, an attempt was made to adapt the samples in cell culture with the intention of achieving the complete sequencing of the genome of circulating rotavirus in rabbits in Mexico, however, its culture was not achieved and to date we have not achieved it, however, the results are as expected since, worldwide it has been reported that the various strains of rotavirus are uncultivable, for this reason we do not include the results in the article, and we hope in the future to be able to adapt some to culture, due to the pandemic by COVID-19 affected optimal development with the restriction placed on Universities in Mexico with access to the facilities.

  1. Could the authors elaborate on the specific environmental or management practices at the affected rabbitry that might have contributed to the outbreak?

Specifically, the farm from which the C3/15 rabbit sample came was a semi-intensive production, located in Amecameca, State of Mexico, at the foot of the Popocatépetl volcano. In this production, the family members were responsible for the care of the animals and it was one of the largest rabbit production units in the area. However, various species, such as sheep, pigs, chickens, were raised in a conventional backyard, there were dogs and birds and even squirrels could enter the facilities. However, these characteristics are not described due to the producers' refusal, since they thought that if their farm was identified with the presence of these diseases, they could have problems with government agencies.

In the same way, it is important to note that due to the pandemic, this farm and many in the region went bankrupt, since their main source of income was the sale of rabbit carcasses for restaurants in the tourist corridor of the area, and due to the closure of these there was no demand and the producers went bankrupt. It is a situation sad, discouraging for many, and invisible because, in the official figures everything is fine and the opposite is not reported, that is why, with these findings we want to support producers so that rabbit farming activity can be resumed and recovered.

This image shows the state of the facilities at the time of sampling.

It is important to mention that as a research group, we provide advice to producers on the implementation of good hygiene and animal management practices, which help them reduce their mortality rates. However, as mentioned, rabbit production is recovering and we will have to wait a little to be able to evaluate the effects and evolution of the findings found prior to COVID-19.

Reviewer 2 Report

Comments and Suggestions for Authors

Reynoso-Utrera et al. conducted an examination of rabbit samples exhibiting enteric symptoms from a rabbit farm in Mexico. The authors utilized specific primers for Rotavirus A (RVA) and detected positive results in 6 out of 188 samples. Genetic analysis of one sample (ID: C-3/15) revealed the G3P[8] genotype, marking the first detection of this particular combination of P and G genotypes in rabbits. Notably, this genotype is genetically closer to human rotaviruses than to those typically found in rabbits.

A key limitation of this study is that the authors analyzed only one of the six positive samples. Consequently, it remains unclear whether the diarrhea in the rabbit farm was caused by rabbit RVA; the infection in rabbit was caused by inter-species infection of human RVA remained unknown。

Major comments

This study is intriguing because it detected a rare genotype, G3P[8], which is closely related to human rotaviruses, in rabbits on a farm. This result suggests that there may have been a minor outbreak of human-type rotavirus at the rabbit farm, or just an accidental infection. It is unfortunate that this hypothesis has not been thoroughly investigated, as the positive samples from six cases were not fully analyzed, which undermines the value of the paper. The authors must analyze at least sequencing of VP4 and VP7 genes of the rest of five samples.

(Optional) If possible, examining the analysis of whole VP4 and VP7 genes and viral isolation and biological characterization are required for more valuable research.

Minor comments

1. Consider to show the nucleotide sequence alignment of C-3/15 and related human RVA strains for discussion whether they are simply genetically or inter-species transmission. This helps to avoid the possibility of laboratory contamination.

2. Lines 105-110. Do the primers used for VP4 and PV7 amplify the genomes of lapine rotavirus strains?

3. Lines 131-135. Do the authors detected 6 of 188 samples and only one sample was subjected to sequencing? How are the other 5 samples?

4. In the phylogenetic trees in Fig.2, please identify the genotypes of each sequences.

5. How was the phylogenetic tree in Figure 4 created? Both VP7 and VP4 nucleotide sequences were involved?

6. Should the strain 7177-1042 shown in Fig. 2  G3 genotype?

7. How was the possibility of the mix infection of multiple genotypes including G3 and P[8] but derived from different strains?

Author Response

Rev 2

Reynoso-Utrera et al. conducted an examination of rabbit samples exhibiting enteric symptoms from a rabbit farm in Mexico. The authors utilized specific primers for Rotavirus A (RVA) and detected positive results in 6 out of 188 samples. Genetic analysis of one sample (ID: C-3/15) revealed the G3P[8] genotype, marking the first detection of this particular combination of P and G genotypes in rabbits. Notably, this genotype is genetically closer to human rotaviruses than to those typically found in rabbits.

A key limitation of this study is that the authors analyzed only one of the six positive samples. Consequently, it remains unclear whether the diarrhea in the rabbit farm was caused by rabbit RVA; the infection in rabbit was caused by inter-species infection of human RVA remained unknown。

In the general context, we agree with reviewer 2 and our main objective is precisely to answer these questions:

Have human RVA strains infected rabbits and caused enteric disease?

Is the infection incidental or is it a common circulating strain?

And finally, and from our point of view, one of the most interesting: is it some kind or new type of reassortment?

However, as has been shown in numerous articles on the identification of rotavirus worldwide, in many cases, the sequence of the various strains has not been obtained and also as found in the reports published worldwide, there is a considerable number of new genotypes reported with the identification of a single sample, with the sequence of the VP4 and VP7 genes, using the same primers and methodology used in this study.

To mention a few:

  • Bányai K, Forgach P, Erdelyi K, Martella V, Bogdan A, Hocsak E, et al. Identification of the novel lapine rotavirus genotype P[22] from an outbreak of enteritis in a Hungarian rabbitry. Virus Res. 2005, 113(2), 73-80.
  • Browning GF, Chalmers RM, Fitzgerald TA, Snodgrass DR. Serological and genomic characterization of L338, a novel equine group A rotavirus G serotype. J Gen Virol. 1991;72 ( Pt 5):1059-1064. doi:10.1099/0022-1317-72-5-1059
  • Bwogi, J., Karamagi, C., Byarugaba, D. K., Tushabe, P., Kiguli, S., Namuwulya, P., Malamba, S. S., Jere, K. C., Desselberger, U., & Iturriza-Gomara, M. (2023). Co-Surveillance of Rotaviruses in Humans and Domestic Animals in Central Uganda Reveals Circulation of Wide Genotype Diversity in the Animals. Viruses15(3), 738.
  • Ghosh S, Kobayashi N, Nagashima S, et al. Full genomic analysis and possible origin of a porcine G12 rotavirus strain RU172. Virus Genes. 2010;40(3):382-388. doi:10.1007/s11262-010-0454-y
  • Guo, D., Liu, J., Lu, Y., Sun, Y., Yuan, D., Jiang, Q., … Qu, L. (2012). Full genomic analysis of rabbit rotavirus G3P[14] strain N5 in China: Identification of a novel VP6 genotype. Infection, Genetics and Evolution, 12(7), 1567–1576.
  • Khamrin P, Maneekarn N, Peerakome S, Yagyu F, Okitsu S, Ushijima H. Molecular characterization of a rare G3P[3] human rotavirus reassortant strain reveals evidence for multiple human-animal interspecies transmissions. J Med Virol. 2006;78(7):986-994. doi:10.1002/jmv.20651
  • Lavazza A, Capucci L, editors. Viral infection of rabbits. Proceedings of the 9th World Rabbit Congress, Verona, Italy, 10-13 Jun 2008.
  • avazza A, Cerioli M, Martella V, Tittarelli C, Grilli G, Brivio R, et al. editors. Rotavirus in diarrheic rabbits: prevalence and characterization of strains in Italian Farms. Proceedings of the 9th World Rabbit Congress, Verona, Italy, 10-11 June 2008.
  • Licois, D, editor. Domestic rabbit enteropathies. Proceedings of the 8th World Rabbit Congress; 2004 Sep 7-10; 2004. Puebla, Mexico. 385-403.
  • Nieddu D, Grilli G, Gelmetti D, Gallazzi D, Toccaciel S, Lavazza A, editors. Electron microscopy detection of viral agents in rabbits with enteropathy during the period 1982-1999 in Italy. Proceedings of the 7th World Rabbit Congress; 2000 July. Valencia, Spain. 4-7.
  • Parreño V, Bok K, Fernandez F, Gomez J. Molecular characterization of the first isolation of rotavirus in guanacos (Lama guanicoe). Arch Virol. 2004;149(12):2465-2471. doi:10.1007/s00705-004-0371-2
  • Perkins C, Mijatovic-Rustempasic S, Ward ML, Cortese MM, Bowen MD. Genomic Characterization of the First Equine-Like G3P[8] Rotavirus Strain Detected in the United States. Genome Announc. 2017;5(47):e01341-17. Published 2017 Nov 22. doi:10.1128/genomeA.01341-17

     Major comments

This study is intriguing because it detected a rare genotype, G3P[8], which is closely related to human rotaviruses, in rabbits on a farm. This result suggests that there may have been a minor outbreak of human-type rotavirus at the rabbit farm, or just an accidental infection. It is unfortunate that this hypothesis has not been thoroughly investigated, as the positive samples from six cases were not fully analyzed, which undermines the value of the paper. The authors must analyze at least sequencing of VP4 and VP7 genes of the rest of five samples.

(Optional) If possible, examining the analysis of whole VP4 and VP7 genes and viral isolation and biological characterization are required for more valuable research.

As previously mentioned, we have indeed neither abandoned nor forgotten this objective and in the region where we work we continue with the study of the presence of rotavirus, attempting to adapt it in cell culture, however during the course of this research it was not possible to adapt it and to date we have not achieved this objective, again without the desire to seem repetitive with the answer, this is not new, as it has been repeatedly mentioned by various research groups in all countries where rotavirus has been identified in different animal species and even in the case of strains found in humans it has been repeatedly cited: “the strains have not been adapted to culture”

In addition to the works included previously, we present the following quote from Damanka, et al., in which they present the identification of a new RV genotype in a child, identified with the same methodology that we present in this work.

Damanka, S., Lartey, B., Agbemabiese, C. et al. Detection of the first G6P[14] human rotavirus strain in an infant with diarrhoea in Ghana. Virol J 13, 183 (2016). https://doi.org/10.1186/s12985-016-0643-y

Minor comments

  1. Consider to show the nucleotide sequence alignment of C-3/15 and related human RVA strains for discussion whether they are simply genetically or inter-species transmission. This helps to avoid the possibility of laboratory contamination.

We have not considered showing this alignment because the phylogenetic trees seem sufficient to us, and at no time has the similarity of the sequences reported in this study been identified with those present in the Rotateq vaccine which was used as a positive control and could be the source of contamination in question, as shown in the Blast results, however if it is considered essential to include it, we can do so.

VP7 (MT267356)

VP4 (MT267357)

  1. Lines 105-110. Do the primers used for VP4 and PV7 amplify the genomes of lapine rotavirus strains?

Yes, these primers are the ones that have been used for the identification of Lapine strains, and they are the ones that are used by consensus for the identification of rotaviruses in different species.

  1. Lines 131-135. Do the authors detected 6 of 188 samples and only one sample was subjected to sequencing? How are the other 5 samples?

All identified samples were processed and attempted to be sequenced, however the sequence was not successful as has been mentioned for the case of rotavirus in other reports, which is why it was not mentioned in the article.

  1. In the phylogenetic trees in Fig.2, please identify the genotypes of each sequences.

Unlike Fig. 1, 3 and 4, where the genotypes of the samples that make up the phylogeny are identified in brackets because they present different genotypes, figure two corresponds exclusively to samples with genotype G3, this clarification has been included in line 178, in addition to what is indicated in the caption of the figure, the suggestion in the image has been addressed.

  1. How was the phylogenetic tree in Figure 4 created? Both VP7 and VP4 nucleotide sequences were involved?

It is correct, for the preparation of figure 4 the combination was included in order to be able to analyze the genotype identified in the sample, in other reports they have only included the analysis of the genotypes separately as identified in figures 1-3, however we find it more enriching to be able to show the complete genotype identified in this study.

  1. Should the strain 7177-1042 shown in Fig. 2 G3 genotype?

That is correct, strain 7177-1042 was erroneously assigned the G4 genotype at the time of its report; in reality it corresponds to a G3 genotype as shown by phylogenetic analyses and as a complementary tool for the assignment of genotypes, RotaC 2.0.

  1. How was the possibility of the mix infection of multiple genotypes including G3 and P[8] but derived from different strains?

It is possible, as mentioned in different studies, infections of multiple rotavirus genotypes have been identified, however, in order to answer this question, as mentioned in the conclusion, it is necessary to continue with the identification of the presence of rotavirus in samples of rabbits from Mexico, obtain their sequences and continue with the investigations to achieve the adaptation of those that are possible to cell culture, and in this way analyze the complete genomic constellation.

Round 2

Reviewer 2 Report

Comments and Suggestions for Authors

I have no further concerns, and I believe the manuscript is now suitable for publication.